# COVIVA: Effect of transcutaneous auricular vagal nerve stimulation on fatigue-syndrome in patients with Long Covid – A placebo-controlled pilot study protocol

**Mortimer Gierthmuehlen**[ID][1]*, **Petra Christine Gierthmuehlen**[2]

**1** Department of Neurosurgery, University Medical Center Knappschaftskrankenhaus Bochum, Ruhr-University Bochum, Bochum, Germany, **2** Department of Prosthodontics, University Medical Center Duesseldorf, Duesseldorf, Germany

\* mortimer.gierthmuehlen@ruhr-uni-bochum.de

## Abstract

*Background*: Up to 80% of patients who develop coronavirus disease-2019 (Covid-19) infection subsequently experience long covid/post-covid syndrome. The World Health Organization (WHO) has estimated that >770 million patients have been infected with Covid-19 globally. Even if only 10% of these patients develop long covid, >75 million patients will suffer for a long period. Among the various symptoms of post-covid syndrome, fatigue is common, affecting up to 60% of the patients. As observed in other viral infections, elevated levels of inflammatory cytokines may play a role. Transcutaneous auricular vagal nerve stimulation (taVNS) is a noninvasive method that modulates the immune system via the central nervous system and has shown promising effects in autoimmune diseases and improving fatigue. In this pilot study, we investigated the feasibility of daily taVNS in patients with long covid-related fatigue. Additionally, the effects of taVNS on fatigue and quality of life will be analyzed. *Methods*: A total of 45 adult patients with long covid associated fatigue syndrome will be enrolled in this study, and will be randomized to the above-threshold-stimulation, below-threshold-stimulation, or sham-stimulation arms, after being informed that they will feel the stimulation. The above-threshold-group will receive a 4-week-long left-sided cymba conchae taVNS with 25 Hz, 250 µs pulse width 28s/32s on/off paradigm for 4 h throughout the day. The below-threshold group will receive stimulation below the sensational threshold, whereas the sham group will receive no stimulation following application of a non-functional electrode. The daily stimulation protocol will be recorded either manually or using the provided app. Three well-established questionnaires, the Multidimensional-Fatigue-Inventory-20, Short-Form-36, and Beck-Depression-Inventory, and the newly established Post-Covid-Syndrome-Score will be completed both before and after 4 weeks of stimulation. *Discussion*: The primary endpoint has been set as the patients' average daily

**Data availability statement:** No datasets were generated or analysed during the current study. All relevant data from this study will be made available upon study completion.

**Funding:** This study was financed by the Department of Neurosurgery at the University Medical Center Knappschaftskrankenhaus Bochum.

**Competing interests:** The authors have read the journal's policy and have the following competing interests: MG is co-founder of Neuroloop GmbH, a company developing an invasive vagal nerve stimulator against arterial hypertension. This does not alter our adherence to PLOS ONE policies on sharing data and materials.

stimulation time after 4 weeks, while secondary endpoints include the effects of taVNS on fatigue and Quality of Live (QoL). As a non-invasive treatment option, taVNS may be a notable alternative for patients with post-covid related fatigue. *Trial registration*: This study was approved by the local ethics committee (23/7798) and registered (DRKS00031974) (see supporting information files). *Ethics & Dissemination:* The ethical justifiability of this study was supported by prior research demonstrating the safety of taVNS. Patients will be recruited by general practitioners, and written informed consent will be obtained. All data will be pseudonymized for collection and storage. The study results will be published in peer-reviewed journals with the aim of providing evidence of the potential of taVNS in long covid management. The study will be conducted in accordance with the principles of the Declaration of Helsinki.

## Introduction

Up to 80% of patients suffer from long-term consequences after recovering from a SARS-CoV2 infection, that hinder their reintegration into daily work and family life [1,2]. These limitations have prompted the recent publication of the German "Standing Guideline Commission of the Association of Scientific Medical Societies in Germany"("Arbeitsgemeinschaft der Wissenschaftlichen Medizinischen Fachgesellschaften" - AWMF) guidelines for the rehabilitation of patients with coronavirus disease-2019 (Covid-19) [3]. In one study, fatigue was observed in 53% of 143 patients [4]. Fatigue is a well-known symptom of viral infections [5] which can significantly impact patients, particularly in patients who have had a Covid-19 infection [6]. This symptom can continue to affect patients for months after recovery from the infection [2,7]. The fatigue severity scale is a frequently used screening tool for fatigue [8], commonly used to diagnose chronic fatigue.

Invasive vagus nerve stimulation (iVNS) has been approved for the treatment of epilepsy and depression since the early the 1990s and 2000s [9]. Vagal activation through vagal nerve stimulation (VNS) leads to immunomodulation, which seems to have a positive effect on autoimmune diseases such as chronic inflammatory bowel disease [10], rheumatoid arthritis [11], and pancreatitis [12] in experimental settings.

Based on the anatomical observation that a specific part of the ear, the cymba conchae, is solely innervated by the vagus nerve [13–15], electrical stimulation of this region has been investigated as a noninvasive alternative to iVNS. Studies have shown that this transcutaneous auricular VNS (taVNS) leads to comparable brain activation as iVNS [16,17]. Furthermore, taVNS has been shown to positively influence epilepsy [18], depression [19], and cytokines in rheumatoid arthritis [20]. Further large-scale studies and compliance with the defined standards [21] are required to further investigate the effectiveness of taVNS under these conditions.

Owing to the pandemic, further research on the causes of fatigue has not yet been conducted, while the underlying causes of fatigue remain inconclusive.

Although conflicting results have been published [22,23], existing research indicates that the development of fatigue is partly triggered by increased levels of pro-inflammatory hormones known as cytokines in the blood, while its severity seems to correlate with the levels of these cytokines in the initial phase of the disease [24]. Studies have further shown that Covid-19 infection leads to a massive increase in cytokines, known as the cytokine storm [25,26], resulting in elevated levels of interleukin (IL) 2, IL-4, and tumor necrosis factor-alpha [27]. Consequently, one Cochrane review demonstrated that IL-6 blocking medication could have a positive impact on the course of acute Covid-19 infection [28]. Overactivation of mast cells also seems to play a possible role in the development of long-term Covid [29], while D-dimer and C-reactive protein levels also appear to correlate with the occurrence of fatigue during the long-term Covid phase [30].

TaVNS has been shown to reduce inflammatory cytokines [31–33], modulate cardiac vagal tone [34], modulate arousal through central pathways [35] and positively influence depressive disorders [36,37], which may play a role in the development of long-covid associated fatigue. This modality has already been applied in pilot studies of patients with acute Covid-19 infection, where it has been shown to help lower cytokine levels [38], thus emphasizing the urgent need for further studies on non-invasive VNS in Covid-19 [39]. The effectiveness of taVNS on fatigue syndromes in systemic lupus erythematosus [40] and Sjögren's syndrome [35,41] was also investigated in a recent clinical trial FatiVa [42]. However, whether taVNS reduces fatigue by reducing inflammatory cytokine levels or through a direct central mechanism remains unclear [43]. However, several studies have discussed the potential use of noninvasive vagal nerve stimulation in the treatment of long-covid, showing promising results [44–47]. The influence of VNS on olfactory function is currently being investigated; however, further research is warranted [48,49].

In summary, vagus nerve stimulation appears to have the potential to reduce the occurrence of fatigue and depression in long covid syndrome by modulating the immunosystem and influencing the central arousal pathways. As reintegration into daily life and work is of paramount importance in patients' quality of life, and the healthcare or retirement system, we aimed to evaluate whether taVNS, which has few side effects, exerts a positive influence on the rehabilitation capacity and acceptance of treatment among long covid patients with fatigue syndrome.

## Materials and methods

### Inclusion criteria:

- History of prior Covid infection within the last 36 months

- Fatigue persisting for ≥ 3 months, with a score ≥4 on the FSS (Fatigue Severity Scale)

- Age > 18 years

- No or stable treatment for depression for at least 4 weeks

### Exclusion criteria:

- Severe psychiatric illness, such as schizophrenia

- Acute Covid infection within the last 14 days

- >8 points on the « Nurses' Global Assessment of Suicide Risk » questionnaire

- Implanted vagus nerve stimulator or history of vagotomy

- Significant heart disease: bradycardia (e.g., sick sinus syndrome), heart failure, or a history of myocardial infarction

- Active implants include pacemakers, defibrillators, neurostimulators, cochlear implants, and drug delivery devices

- Inability to understand the study plan

- Progressive neurological disease (e.g., Parkinson's, Multiple Sclerosis, epilepsy)

- Pregnancy

- Other acute illnesses or conditions associated with fatigue (e.g., cancer, chemotherapy treatment, and autoimmune diseases)

- Polyneuropathy

- Presence of skin conditions, such as infection, psoriasis, or eczema in the treatment area

- Presence of any anatomical anomaly that prevents successful placement of the ear electrode

- Presence of any serious illness that prevents successful participation in the study.

### Questionnaires

Three well-established questionnaires will be used: the ShortForm36 (SF36), the Beck Depression Inventory (BDI), and the Multidimensional Fatigue Inventory (MFI20). In addition, the newly established Post-Covid-Syndrome-Score (PCS Score) of the "National Pandemic Cohort Network" Consortium [50] will be included.

### Abortion criteria

Patients will discontinue participation if they present with any of the exclusion criteria.

### Endpoints

The present study aims to investigate whether taVNS can influence fatigue and the associated quality of life in patients with long covid. To achieve this, the following endpoints will be evaluated:

- The primary endpoint is acceptance of taVNS in patients with a long stimulation duration, measured by the average daily stimulation duration.

- Secondary endpoints include the effects of taVNS on the MFI20, BDI, quality of life (SF36), and PCS-Score.

### Selection of patients and data collection

A total of 45 patients will be recruited and divided into three groups. Patients will be informed about the study through flyers and by their general practitioners, and will be provided with access to a website. Additionally, local support groups will be contacted and informed. On the website, questions regarding the inclusion and exclusion criteria will be asked anonymously, without data storage. At the end of the questionnaire, patients will be informed of whether they qualified for the study. In the case of study qualifications, patients will be provided with an email address where they can sign up to participate. Patients will be informed about the study aims and procedures by an investigator or certified medical professional. After obtaining written informed consent, data collection will be pseudonymized in the RedCap electronic case report form (eCRF) system of Ruhr-University-Bochum and will be General Data Protection Regulation (GDPR)-compliant. The RedCap system automatically randomizes patients into Verum-1, Verum-2, or Sham groups (n = 15 each). The patients will be instructed on how to use the simulator for 4 h per day, with at least 1 h of consecutive stimulation in the left ear. Patients will be asked to log all stimulation times manually in a diary. Because patients in the sham group will also receive the device's user manual, which states that they should feel a tingling sensation in the ear, they will be instructed to use the highest stimulation intensity that is not perceptible to them. After adjusting the stimulation to this level, sham patients will receive a non-functional electrode that does not perform any stimulation. This approach is justified as the principle "more

is better" does not apply to vagus nerve stimulation, and stimulation below the perception threshold could still, theoretically, have an effect.

## Patient and public involvement statement

Local support groups, regional long-term outpatient clinics, and general practitioners all participated in the study design. They were informed that the study would take place, and encouraged to disseminate this information among their patients and study flyers.

## Sample size calculation

The group size was based on the aforementioned studies on taVNS in other forms of fatigue [31,35,41], our aforementioned FatiVa study [42], and an ongoing similar study in the USA which enrolled 40 patients (NCT05630040). Recommendations for group size made in previous pilot studies were also considered [51,52]. Based on the expectation of the superiority of the two target groups (low- and high-threshold stimulation) of at least 5 and 10 MFI units, respectively, compared to the placebo (shift), and a type I error of alpha = 0.05, a total of $3 \times 12$ cases = 36 cases were deemed to be required to achieve a study power of 80% for the most important secondary endpoint, the MFI20 questionnaire. The mentioned numbers were secured with a safety margin of approximately 10%–20% (depending on the dropout rate), resulting in a total of $3 \times 15$ cases = 45 cases, if necessary. The calculation will be performed using the R Package "MultNonParam" [53].

## Data analysis

The two-sided p-value of Fisher's exact test will be used to evaluate the difference between the three groups (primary outcome is an incidence rate with patients who exceed a usage-time-threshold). Additionally, the two treatment groups will be descriptively compared with the placebo group, and between themselves. The rationale to use Fisher's exact test is to compare binary data between groups (in total three two-group comparisons). The rationale for the Kruskal-Wallis rank analysis of variance is to avoid the potential influence of outliers and to handle unknown data distributions (normally distributed data are not expected).

Since three tests (each group compared with each other) are performed and the threshold for the p-value was set at an overall (studywise) alpha of 0.05, then three tests result in an Bonferroni-adjusted (comparisonwise) threshold of p/3 = 0.017. p-values below this threshold denote a statistical significant difference.

Secondary outcome measures included the MFI20, BDI, SF-36, and PCS-score, which were modified similarly to the primary outcome measures. BDI, SF36 and PCSS are other secondary measures which will be analyzed descriptively. Since those parameters are continous rank-analses of variance (Kruskal wallis tests) will be used. Descriptive statistics, such as means, medians, quantiles, and respective case numbers of the available data, will be reported. The between-group contrast will be reported as the mean difference with 95% confidence intervals. Additionally, a nonparametric estimation of the group effect (shift) using the Hodges-Lehmann estimator shift will be provided.

Successful acceptance and compliance will be defined as achieved when the acceptance criteria (average taVNS for 4h/day) are met by at least 80% of patients. A total of 45 patients will be enrolled, with 15 assigned to each of the three groups: "sham", "Verum-1", and "Verum-2." The decision to have 15 patients per group was based on feasibility, as well as from influence by studies that examined noninvasive VNS for sleep deprivation, lupus, and Sjoegren's syndrome. Furthermore, the selected sample size was aligned with the recommended sample sizes for feasibility studies [51,52].

## Study plan (Fig 1)

**Verum stimulation.** This study included two verum-stimulation groups. In both groups, taVNS will be administered according to the device approval at the cymba conchae of the left ear. Because establishing a placebo group in stimulation studies is generally difficult, two Verum groups were planned: one will receive stimulation above the sensory

| | | Study Period | | |
|---|---|---|---|---|
| | Enrolment | Allocation | Close-out | *Optional:* |
| **TimePoint** | -t1 | Visit 0 | Visit 1 | Visit 2 |
| **Enrolment** | | | | |
| **Eligibility screen (via Website, mail or phone)** | x | | | |
| **Informed consent** | | x | | |
| **Post-Covid-Severity-Score** | | x | | |
| **Allocation** | | x | | |
| **Interventions:** | | | | |
| **VERUM 2: Stimulation (25Hz, 250µs, 28s on / 32s off) above threshold** | | START | END | |
| **VERUM 1: Stimulation (25Hz, 250µs, 28s on / 32s off) below threshold** | | START | END | |
| **SHAM: Sham-Stimulation (non-functional fake-electrode)** | | START | END | |
| **Optional for participants of VERUM 1 and SHAM-Groups: VERUM 2-Stimulation** | | | START | END |
| **Assessments:** | | | | |
| **Baseline Short-Form 36** | | x | | |
| **Baseline Beck-Depression-Inventory** | | x | | |
| **Baseline Multidimensional-Fatigue-Inventory** | | x | | |
| **Baseline Post-Covid-Severity-Score** | | x | | |
| **Outcome Short-Form 36** | | | x | |
| **Outcome Beck-Depression-Inventory** | | | x | |
| **Outcome Multidimensional-Fatigue-Inventory** | | | x | |
| **Outcome Post-Covid-Severity-Score** | | | x | |

**Fig 1. SPIRIT schedule of enrollment of the CoViva Study.**

threshold, and one below. If subthreshold stimulation also leads to a positive effect, future studies may omit the suprathreshold stimulation group.

Stimulation will be conducted using the commonly chosen parameters in the literature, which have been successfully applied over a period of 20 weeks [54]. These parameters include a frequency of 25 Hz and a pulse width of 250 µs. These stimulation parameters will be preset in the commercially available, CE (Conformité Européenne, European equivalent to FDA) tVNS-L stimulator and cannot be changed. Stimulation will be applied throughout the day for 4 h using a 28s on/ 32s on/off schedule. Patients will be asked to download available Android and iOS apps, which will be paired with the stimulator via Bluetooth to document the daily stimulation time. Based on these records, patients will be able track when the recommended 4 h daily stimulation target day is reached. Although the stimulator will be turned off after 4 h of continuous stimulation, there is no automatic blocking of the stimulator after 4 h of cumulative stimulation, as only patients who understood the study protocol will participate, in accordance with the exclusion criteria. Additional negative effects of a stimulation lasting more than 4 hours have not been reported in literature – however, the likelihood of known side effects such as headache, ear pain, skin irritation or nausea [55] increases if stimulation lasts longer than 60min [56].

**Verum-1 stimulation.** The intensity will initially be set to produce a mild tingling sensation in the ear that should not be painful. Subsequently, the intensity will be reduced to a level at which the patient no longer feels the stimulation. The patient will then be asked to repeat this procedure every time stimulation is initiated to ensure that the stimulation is always subthreshold.

**Verum-2 stimulation.** The intensity will be adjusted according to the instructions provided to produce a mild, non-painful tingling sensation in the ear. The patients will be able modify the stimulation intensity during the study to ensure that it remains above this threshold.

**Sham stimulation.** The intensity will initially be set to produce a mild tingling sensation in the ear that should not be painful. Subsequently, the intensity will be reduced to a level at which the patient no longer feels any stimulation. The stimulator will be maintained at this level throughout the study. However, participants in the sham group will receive an ear electrode provided by the manufacturer, which is based on a regular electrode but does not establish an electrical connection between the stimulator and ear contact, ensuring no actual stimulation occurs.

### Study procedure

The study commenced on 1st September 2023. As there is no central point of contact for patients with long covid, all participants were/will be informed about the study by their general practitioners. Contact will be established between general practitioner networks and local support groups. General practitioners can identify suitable patients based on the inclusion and exclusion criteria, and referred them to the study investigator. The two visits ("Visit 0" and "Visit 1") and the telephone interview will subsequently be scheduled following consultation with the patient.

During the study period, participants will use the "tVNS Patient" app developed by tVNS Technologies GmbH (available for free in the respective app stores) to automatically document the frequency and duration of stimulations. Compliance documentation will be provided, as recommended in the 2020 consensus paper on noninvasive VNS [21]. Only the patient can access these data, which will be finally reviewed by a physician at the end of the stimulation period. All data will be stored exclusively on participants' smartphones. The questionnaires will be entered and analyzed as electronic case report forms (eCRFs) in the RedCap system of the Department of Medical Informatics.

**Visit 0.** During the initial visit, patients will receive informational documents, sign informed consent forms, and complete the questionnaires. Personal data such as age, sex, height, weight, therapy time, duration, and type will also be documented. The patients will receive the stimulator and be instructed regarding its use by the investigator, according to the guidelines of the German Medical Devices Act (MPG/MPDG). Additionally, group assignment (SHAM, Verum 1, or Verum 2) will be performed. Subsequently, the 4-week stimulation phase will be initiated.

**Telephone interview after 7 days.** After 7 days, patients will be contacted by phone or email (if preferred) to inquire about their use of the stimulator, any side effects, or any potential questions they may have.

**Visit 1** After 4 weeks, the patients will complete the questionnaires again. Personal data, including weight, will be collected, while data regarding the duration of stimulation will be retrieved from a smartphone. Subsequently, the stimulators will be returned. Patients who cannot attend visit 1 because of their health conditions will receive online access to the questionnaire section of the eCRF to complete it from home.

**Optional Visit 2 (optional cross-over-design).** Patients enrolled in "Sham" or "Verum 1" will be offered to receive "Verum 2" stimulation for 4 weeks at the end of the study.

**The study will be terminated here.**

## Discussion

### Safety

Several studies have shown that healthy patients undergo acute taVNS to investigate scientific questions in behavioral research [57–59]. Although mild side effects may occur, none of the existing studies on this topic have reported any severe side effects. TaVNS was also investigated in a medical doctoral thesis involving healthy patients without relevant side effects [60]. In total, the cited studies included >1,200 healthy patients who underwent taVNS without any difficulty. Although the occurrence of side effects correlates with the duration of stimulation [61], even during chronic taVNS, which lasts several weeks, only a few adverse effects and no severe events have been observed [13,56,62].

A recent review of transcutaneous VNS (including both cervical and auricular stimulations) reported skin redness (16.7%), headache (3.3%), nasopharyngitis (1.6%), and nausea/dizziness (1.1%) as the most common side effects

among 1,322 patients. Only 2.6% of all patients had to discontinue the study because of side effects, while facial palsy occurred only with cervical vagus nerve stimulation [55]. Long covid and its associated fatigue syndrome pose a challenging burden for both patients and healthcare systems. Investigating a noninvasive method to improve this devastating condition may be extremely beneficial, and this goal could justify these known side effects. The results of this pilot study will be analyzed with respect to subsequently conducting a full multicenter interventional study, and will provide information regarding future inclusion and exclusion criteria, recruitment rate, sample size, expected adherence, and dropout rates among patients with long covid-associated fatigue.

## Ethics and regulatory aspects

This study was approved by the local ethics committee of Ruhr-University-Bochum under registration number 23/7798 on 05/12/23, with amendment acceptance on 06/14/23. It was further registered in the DRKS-System under the number DRKS00031974 (https://drks.de/search/de/trial/DRKS00031974) on 05/25/23. The accepted version was version 3.0, from 06/14/23, and the sponsor was the University Medical Center Knappschaftskrankenhaus Bochum. The study will be performed under the §47(3) of the new MPDG law defining clinical studies involving medical devices. As the tVNS-L-System, owing to the fixed set of stimulation parameters, possesses a CE mark for vagal nerve stimulation, and this study involves no invasive procedures, ethical approval was relatively simple and rapid to obtain. Thus, if the results indicate feasibility, a larger study could be quickly initiated with a CE-certified system, in contrast to a system that does not have a CE mark.

## Sham intervention

Defining the sham group in a stimulation study is challenging. Several different approaches have been discussed in the literature, including stimulation of the ear lobe instead of the cymba conchae, or stimulation of other parts of the body [63–66]. However, an informed patient may realize that such stimulation is different from normal taVNS, and could thus identify themselves as being in the sham group. To address this, patients will alternately provide with information that they will either feel or do not feel the stimulation. Once the tingling sensation has been demonstrated to the patient, the intensity will be reduced until the patient does not feel any stimulus. The sham group will then be provided with a non-functional electrode, while the verum-1-group will maintain their functional electrode throughout the study phase. If the analysis shows that both the verum-1 and verum-2 interventions have a significant effect on fatigue, future studies will need only one subthreshold group and one sham group, leaving the patients blinded to which group they have been included in.

Within this feasibility trial, no invasive procedures, such as blood sampling, has been planned, in order to maintain the regulatory requirements as low as possible. The stimulation system tVNS-L® is manufactured by tVNS GmbH in Erlangen, Germany. This system allows modification of the stimulation intensity but not of the pulse width or frequency. As such, it is a CE-mark for transcutaneous auricular vagal nerve stimulation. The variant tVNS-R® (R for research) allows modification of all parameters; consequently, it does not have a CE-mark, making it extremely difficult to obtain insurance and ethical approval for a scientific study.

The results of this study will be published in scientific, probably open-access, journals with a scope for public health. If taVNS, as a very safe procedure, can improve fatigue and the challenging sequelae of long covid, it would significantly improve the quality of life of such patients. This potential improvement in the quality of life justifies the extremely low risk of taVNS therapy in this study.

## Data protection, management, and monitoring

Only pseudonymized data will be stored on the RedCap®-System in our hospital, while a table containing patient records will be stored on a different drive within the hospital network. Only the investigator and study nurses will have access to the database, and the rules of the GDPR will be applied. Data monitoring will be performed by the Department of Medical

Informatics, Biometry, and Epidemiology at Ruhr-University Bochum, which runs and maintains the RedCap system. The results of the study will be made available in peer reviewed journals.

## Supporting information

**S1 SPIRIT-Checklist.**
(XLSX)

## Acknowledgments

Not applicable.

## Author contributions

**Conceptualization:** Mortimer Gierthmuehlen, Petra Christine Gierthmuehlen.

**Methodology:** Mortimer Gierthmuehlen, Petra Christine Gierthmuehlen.

**Project administration:** Mortimer Gierthmuehlen.

**Supervision:** Mortimer Gierthmuehlen, Petra Christine Gierthmuehlen.

**Validation:** Mortimer Gierthmuehlen.

**Writing – original draft:** Mortimer Gierthmuehlen, Petra Christine Gierthmuehlen.

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
