## [Decision Letter · Decision Letter 0]

10 Jul 2024

PONE-D-23-38498COVIVA: Effect of transcutaneous auricular vagal nerve stimulation on the fatigue-syndrome in patients with Long Covid – a placebo-controlled pilot study protocolPLOS ONE

Dear Dr. Gierthmuehlen,

Thank you for submitting your manuscript to PLOS ONE. After careful consideration, we feel that it has merit but does not fully meet PLOS ONE’s publication criteria as it currently stands. Therefore, we invite you to submit a revised version of the manuscript that addresses the points raised during the review process.

Two reviewers have evaluated the manuscript. Reviewer #2, in particular, raised several major concerns regarding the methods used in the paper.

We look forward to receiving your revised manuscript.

Kind regards,

Mohammadreza Pourahmadi, PT, Ph.D., Postdoctoral Fellow

Academic Editor

PLOS ONE

Journal Requirements:

3. Thank you for stating the following in the Competing Interests section: "MG is founder and advisor of Neuroloop GmbH, a start-up investigating invasive vagal nerve stimulation against arterial hypertension. PG received grants from Ivoclar Vivadent, Straumann, Vita Zahnfabrik und Deutsche Forschungsgemeinschaft DFG."

We note that one or more of the authors are employed by a commercial company: Neuroloop GmbH

Reviewers' comments:

Reviewer's Responses to Questions

**Comments to the Author**

1. Does the manuscript provide a valid rationale for the proposed study, with clearly identified and justified research questions?

Reviewer #1: Yes

Reviewer #2: Partly

2. Is the protocol technically sound and planned in a manner that will lead to a meaningful outcome and allow testing the stated hypotheses?

Reviewer #1: Partly

Reviewer #2: Yes

3. Is the methodology feasible and described in sufficient detail to allow the work to be replicable?

Reviewer #1: Yes

Reviewer #2: Yes

4. Have the authors described where all data underlying the findings will be made available when the study is complete?

Reviewer #1: Yes

Reviewer #2: No

5. Is the manuscript presented in an intelligible fashion and written in standard English?

Reviewer #1: Yes

Reviewer #2: Yes

6. Review Comments to the Author

You may also provide optional suggestions and comments to authors that they might find helpful in planning their study.

Reviewer #1: Thank you for the opportunity to review this paper. This is an interesting manuscript presenting a protocol of placebo-controlled pilot study on effect of transcutaneous auricular vagal nerve stimulation on the fatigue syndrome. The primary objective of this study is to investigate the effect of taVNS on patients with long Covid fatigue syndrome.

My review mainly concerns only the statistical aspects of the study. Some questions reported below were raised and in my view, it is not acceptable in this form for the publication in this journal.

1. The sample size justification reported is incomplete in relation to primary outcame. Is not indicate the primary end-point variable on which the estimation is evaluated; the common SD estimated is not reported;

2. In the statistical analysis paragraph should be more detailed the model applied for the analysis with relative post-hoc analysis;

3. There are some error of typing. Please, verify the manuscript for English and typos.

Reviewer #2: Summary

This paper presents the COVIVA study protocol, which aims to evaluate the feasibility of transcutaneous auricular vagal nerve stimulation (taVNS) as a treatment for patients with long covid syndrome and its potential to alleviate fatigue. The description of the planned study is informative but requires additional details and justification to ensure reproducibility. For example, the statistical test employed for the sample size calculation and the theoretical effect size underlying this calculation need to be clearly outlined. The findings of this study have the potential to guide the development of taVNS as a treatment for patients with Long-COVID syndrome. Therefore, this protocol is worthy of publication, provided that the methods and materials are thoroughly revised and my major comments are adequately addressed.

Major comments

1. The theoretical foundation of developing taVNS as a treatment for covid- associated fatigue in this study is its anti-inflammatory effect. While the anti-inflammatory effect of taVNS is well-established, it remains unclear whether taVNS alleviates fatigue through the anti-inflammatory reflex. To address this, researchers must examine the relationship between fatigue and inflammatory cytokines, which is not the case in reference [15], [35] and [36]. Note that reference [10] states that elevated IL-10 and IL-6 are important at the early stage but do not affect persistent fatigue once the fatigue syndrome is established. Another study has attempted to investigate whether taVNS improves fatigue through modulating inflammatory markers but found no correlation between fatigue and proinflammatory cytokines (please take a look at their results and discussion https://www.neuromodulationjournal.org/article/S1094-7159(22)01263-6/fulltext ) They stated “it has been established that VNS including nVNS modulates EEG microstates and brain activity”, suggesting VNS improving fatigue might be a central mechanism. This is supported by a behavioral and physiological study of the effect of taVNS on fatigue (https://www.brainstimjrnl.com/article/S1935-861X(24)00060-3/fulltext ), which found that the taVNS modulates the arousal pathway and rescue fatigue at the later stage of memory tasks. This is a more direct evidence to support the use of taVNS in patients with long-covid syndrome in comparison to studies about its anti-inflammatory effects.

In the current manuscript, the link between fatigue and inflammation after COVID-19 infection and the anti-inflammation effects of taVNS is missing. I suggest that the author introduces taVNS as a potential intervention after line 84, The introduction of taVNS should include its history (current line 98 - 110) and current knowledge about its positive effects on fatigue such as the results from the aforementioned studies. The authors should then elaborate on the potential mechanism of fatigue and the mechanism of taVNS. Please verify if the inflammation persist months after the infection of COVID-19 potentially leading to fatigue because intuitively, the inflammation might disappear when the patient recovered from COVID-19. Again, based on the current literature, the effect of taVNS on neuroepinepherine is more likely to be the mechanism for its positive effect on fatigue, if there is any.

The authors state, “Due to the current pandemic, further research on the causes of fatigue has not yet been conducted, and the underlying cause of fatigue remains inconclusive.” The current flow of the introduction implies that this protocol will test whether taVNS can mitigate fatigue through the regulation of pro-inflammatory cytokines, which is not the case. Please restructure the introduction to focus more on the effects of taVNS on fatigue and its feasibility. For instance, what is the significance of establishing the feasibility of taVNS in patients with long-COVID syndrome? Does taVNS need to be administered for a specific duration to treat fatigue?

Please avoid statements such as “In summary, vagus nerve stimulation appears to have the potential to reduce the occurrence of fatigue and depression in long Covid syndrome through the reduction of cytokines, or treat these symptoms.”

2. More details about the methods need to be provided. For example, Line 130. Fatigue persisting for >= 3 months. Is it a subjective report of fatigue? Please introduce the fatigue scale in the introduction. In the exclusion criteria, how to define acute Covid infection. 14 days within the infection? What is Vagus nerve stimulation (line 139) in the exclusion criteria? Do you mean having done a surgical procedure for implanting the cervical VNS device? Please make the inclusion/exclusion criteria as objective as possible. I suggest consolidating Questionnaires: and endpoints (line 167) into one section: primary outcomes.

One of the outcomes is average stimulation duration. Is the total stimulation duration controlled? Will the patient receive instructions about the stimulation duration? Second outcome is MF120, as this is the metric that the author will use to calculate sample sizes. Please clearly state that analysis of other questionnaires is considered exploratory analysis, and the exploratory result should not be used to draw conclusion once the study is completed.

The sample size calculation needs to be more specific. To calculate the sample size, one will need to define the test that will be used, the power that needs to be reached, and the theoretical effect size. Line 202 “The group size is based on the aforementioned studies on taVNS in other forms of fatigue 202 (29,36,37), our aforementioned FatiVa study (38),” Simply using the number of patients of previous studies is not a scientific way for defining sample size because previous studies can be under-powered. “The threshold for the p-value is an alpha of 0.05,” which can be called the significance level.

Line 213 “Based on the expectation of a superiority

213 of the two target groups (low-threshold and high-threshold stimulation) of at least 5 and 10 MFI

214 units, respectively, compared to placebo (shift), and a type I error of alpha = 0.05, a total of 3

215 x 12 cases = 36 cases are required to achieve a study power of 80% in the most important

216 secondary endpoint, the MFI20 questionnaire. The” What is the statistic test that will be used? Is there any evidence for the expectation of 5-10 MFI units?

Could you explain what successful acceptance is and how it will be used? “Successful acceptance and compliance are achieved when the acceptance criteria, as defined

228 in the primary endpoint, are met by at least 80% of patients.” What is the acceptance criteria? Do you mean if more than 80% of patients use taVNS for more than 4 hours, then taVNS is feasible?

Line 264, it seems that this study includes a titration of intensity. Please state how the investigator will reduce the intensity from the initial setting, for example, with a decrement of 5 units when the patient reports feeling the stimulation. How do you ensure that the Verum -1 group uses the predefined stimulation intensity?

In study plan, the study started last year. Have any patients been recruited? Is there any preliminary data that can be used to estimate the effect size?

3. In conflict of interest, as “MG is founder and advisor of Neuroloop GmbH,” The authors need to state whether this company will manufacture products to reduce fatigue in patients with long-covid syndrome.

4. In the discussion section, fatigue is stated as one of the side effects of taVNS in a meta-analysis published in a scientific report. Please discuss whether this side effect will confound the analysis.

5. Please verify if cymba conchae, rather than tragus, is most innervated by the auricular branch of the vagus nerve. (https://www.ncbi.nlm.nih.gov/pmc/articles/PMC7083568/). In reference [22], it seems that they also indicates that tragus is partly innervated by the auricular branch of the vagus nerve. The authors might need additional evidence to support stimulation of aVN through tragus, for example, tragus is easier to access.

Minor comments

1. Abstract, first sentence, I understand this data comes from study. But to provide a background for this protocol, I suggest that the authors state for example, estimated number of patients who is currently having long covid syndrome.

2. Page 3, line 70, “In addition to fatigue (58%), these patients experience headaches

(44%), anosmia, anxiety (13%), and depression (12%). Please specify the denominator used to calculate these percentages.

3. Key message, page 3, is the key message section required to be published in PLOS ONE? Key messages in the manuscript seem to be a shorter abstract. The manuscript without the key messages is complete.

4. Please rephrase sentences 81 – 83; for example, ‘fatigue can continue to affect these patients months after recovery from the infection.’

5. The use of abbreviations needs to be consistent. For example, line 115 is transcutaneous VNS taVNS?

6. What does Verum stand for?

7. Change references into English. Are the journal name and publication date in Germany?

8. Optional: provide instruction to the three groups of patients as supplementary material.

9. Lin 240, please revise “the effect between 239 suprathreshold and subthreshold

10. 240 stimulation will be investigated using two Verum groups.”

11. What is CE in “CE-certified” Line 247

12. In PLOS ONE protocol guideline, the authors would need to state how data saturation will be determined. Also, data availability statement is missing, please check PLOS ONE DATA policy.

13. I recommend registering this study on a research such as OSF.

7. PLOS authors have the option to publish the peer review history of their article (what does this mean? ). If published, this will include your full peer review and any attached files.

**Do you want your identity to be public for this peer review?** For information about this choice, including consent withdrawal, please see our Privacy Policy .

Reviewer #1: No

Reviewer #2: **Yes: ** Gansheng Tan

---

## [Author Response · Author response to Decision Letter 0]

31 Aug 2024

We thank the reviewers and the editor for the interesting and thoughtful comments and especially the mentioned internet resource OSF.

Editor:

When completing the data availability statement of the submission form, you indicated that you will make your data available on acceptance. We strongly recommend all authors decide on a data sharing plan before acceptance, as the process can be lengthy and hold up publication timelines. Please note that, though access restrictions are acceptable now, your entire data will need to be made freely accessible if your manuscript is accepted for publication. This policy applies to all data except where public deposition would breach compliance with the protocol approved by your research ethics board. If you are unable to adhere to our open data policy, please kindly revise your statement to explain your reasoning and we will seek the editor's input on an exemption. Please be assured that, once you have provided your new statement, the assessment of your exemption will not hold up the peer review process.

This is only a study protocol, so there is no additional data which could be published at this point. Once the study is completed and analyzed, the results will be made available within the publication of the results.

Reviewer #1: Thank you for the opportunity to review this paper. This is an interesting manuscript presenting a protocol of placebo-controlled pilot study on effect of transcutaneous auricular vagal nerve stimulation on the fatigue syndrome. The primary objective of this study is to investigate the effect of taVNS on patients with long Covid fatigue syndrome.

My review mainly concerns only the statistical aspects of the study. Some questions reported below were raised and in my view, it is not acceptable in this form for the publication in this journal.

1. The sample size justification reported is incomplete in relation to primary outcame. Is not indicate the primary end-point variable on which the estimation is evaluated; the common SD estimated is not reported;

Statistically, the difference between the two groups is checked in continuous data (main target parameters and ranked judgment scales) using exact rank variance analysis according to Kruskal and Wallis (R package “coin”) to produce a 2-sided p-value. The threshold for the p-value is an alpha of 0.05, i.e. if the test value falls below the threshold, the null hypothesis can be rejected and a significant general difference between the three groups (placebo, low-threshold, higher-threshold group) can be assumed. In addition, the two treatment groups are each compared descriptively with placebo (and with each other). Secondary target parameters are the BDI, the SF-36 and acceptance; these are upgraded in analogy to the main target parameter. Since exactly one main target parameter (MFI20 change) is evaluated for confirmatory purposes, no adjustment is necessary Type 1 risk is necessary. Based on the expectation of superiority of the two target groups (low and higher threshold stimulation) of at least 5 or 10 MFI units compared to placebo (shift) and a type 1 error of alpha = 0.05, a total of 3 x 12 cases = 36 cases in total needed to achieve a study power of 80%. The numbers mentioned are minimum numbers and should be secured by a safety surcharge (depending on the drop-out rate) of around 10% - 20%, so that (if the surcharge is necessary) this results in a total of 3 x 15 cases = 45 cases in total. The calculation was carried out using the R package “MultNonParam” (Kolassa & Seifu, 2013). The remaining parameters (BDI, SF-36, acceptance judgment) are evaluated purely descriptively at the threshold of p = 0.05 (same statistical test as the main target parameter ). The calculated p-values are always two-tailed. Means, medians, quantiles and the respective sample numbers of the available data are given descriptively. For the between-group contrast, the mean difference and its 95% confidence intervals are given as the effect measure. In addition, a nonparametric estimate of the group effect (shift) using the Hodges-Lehman estimator shift (Lehmann, 1998) is given

2. In the statistical analysis paragraph should be more detailed the model applied for the analysis with relative post-hoc analysis;

Since there is a single primary target parameter the confirmatory analysis will not be adjusted for multiple testing. All other parameters will be tested descriptively only (generating hypotheses only).

3. There are some error of typing. Please, verify the manuscript for English and typos.

We had the manuscript proofread.

Reviewer #2: Summary

This paper presents the COVIVA study protocol, which aims to evaluate the feasibility of transcutaneous auricular vagal nerve stimulation (taVNS) as a treatment for patients with long covid syndrome and its potential to alleviate fatigue. The description of the planned study is informative but requires additional details and justification to ensure reproducibility. For example, the statistical test employed for the sample size calculation and the theoretical effect size underlying this calculation need to be clearly outlined. The findings of this study have the potential to guide the development of taVNS as a treatment for patients with Long-COVID syndrome. Therefore, this protocol is worthy of publication, provided that the methods and materials are thoroughly revised and my major comments are adequately addressed.

Major comments

1. The theoretical foundation of developing taVNS as a treatment for covid- associated fatigue in this study is its anti-inflammatory effect. While the anti-inflammatory effect of taVNS is well-established, it remains unclear whether taVNS alleviates fatigue through the anti-inflammatory reflex. To address this, researchers must examine the relationship between fatigue and inflammatory cytokines, which is not the case in reference [15], [35] and [36]. Note that reference [10] states that elevated IL-10 and IL-6 are important at the early stage but do not affect persistent fatigue once the fatigue syndrome is established. Another study has attempted to investigate whether taVNS improves fatigue through modulating inflammatory markers but found no correlation between fatigue and proinflammatory cytokines (please take a look at their results and discussion https://www.neuromodulationjournal.org/article/S1094-7159(22)01263-6/fulltext ) They stated “it has been established that VNS including nVNS modulates EEG microstates and brain activity”, suggesting VNS improving fatigue might be a central mechanism. This is supported by a behavioral and physiological study of the effect of taVNS on fatigue (https://www.brainstimjrnl.com/article/S1935-861X(24)00060-3/fulltext), which found that the taVNS modulates the arousal pathway and rescue fatigue at the later stage of memory tasks. This is a more direct evidence to support the use of taVNS in patients with long-covid syndrome in comparison to studies about its anti-inflammatory effects.

We corrected this accordingly and added respective notes and explanations.

In the current manuscript, the link between fatigue and inflammation after COVID-19 infection and the anti-inflammation effects of taVNS is missing. I suggest that the author introduces taVNS as a potential intervention after line 84, The introduction of taVNS should include its history (current line 98 - 110) and current knowledge about its positive effects on fatigue such as the results from the aforementioned studies. The authors should then elaborate on the potential mechanism of fatigue and the mechanism of taVNS. Please verify if the inflammation persist months after the infection of COVID-19 potentially leading to fatigue because intuitively, the inflammation might disappear when the patient recovered from COVID-19. Again, based on the current literature, the effect of taVNS on neuroepinepherine is more likely to be the mechanism for its positive effect on fatigue, if there is any.

We corrected this accordingly.

The authors state, “Due to the current pandemic, further research on the causes of fatigue has not yet been conducted, and the underlying cause of fatigue remains inconclusive.” The current flow of the introduction implies that this protocol will test whether taVNS can mitigate fatigue through the regulation of pro-inflammatory cytokines, which is not the case. Please restructure the introduction to focus more on the effects of taVNS on fatigue and its feasibility. For instance, what is the significance of establishing the feasibility of taVNS in patients with long-COVID syndrome? Does taVNS need to be administered for a specific duration to treat fatigue?

The reason to investigate feasibility is to find out whether patients with severe fatigue are even able to use taVNS for 4 hours per day. Our experience with the first patients is that having something to do for 4 hours can be a burden for these patients.

Please avoid statements such as “In summary, vagus nerve stimulation appears to have the potential to reduce the occurrence of fatigue and depression in long Covid syndrome through the reduction of cytokines, or treat these symptoms.”

We corrected this.

2. More details about the methods need to be provided. For example, Line 130. Fatigue persisting for >= 3 months. Is it a subjective report of fatigue? Please introduce the fatigue scale in the introduction. In the exclusion criteria, how to define acute Covid infection. 14 days within the infection? What is Vagus nerve stimulation (line 139) in the exclusion criteria? Do you mean having done a surgical procedure for implanting the cervical VNS device? Please make the inclusion/exclusion criteria as objective as possible. I suggest consolidating Questionnaires: and endpoints (line 167) into one section: primary outcomes.

We added the scale and the respective explanatations as suggested. Our ethics committee required us to define the acceptance rate as the primary outcome and the questionnaires as the secondary outcome.

One of the outcomes is average stimulation duration. Is the total stimulation duration controlled? Will the patient receive instructions about the stimulation duration? Second outcome is MF120, as this is the metric that the author will use to calculate sample sizes. Please clearly state that analysis of other questionnaires is considered exploratory analysis, and the exploratory result should not be used to draw conclusion once the study is completed.

The patients will be instructed, we added a respective paragraph. MFI20 is a secondary outcome – the primary outcome is acceptance.

The sample size calculation needs to be more specific. To calculate the sample size, one will need to define the test that will be used, the power that needs to be reached, and the theoretical effect size. Line 202 “The group size is based on the aforementioned studies on taVNS in other forms of fatigue 202 (29,36,37), our aforementioned FatiVa study (38),” Simply using the number of patients of previous studies is not a scientific way for defining sample size because previous studies can be under-powered. “The threshold for the p-value is an alpha of 0.05,” which can be called the significance level.

Line 213 “Based on the expectation of a superiority

213 of the two target groups (low-threshold and high-threshold stimulation) of at least 5 and 10 MFI

214 units, respectively, compared to placebo (shift), and a type I error of alpha = 0.05, a total of 3

215 x 12 cases = 36 cases are required to achieve a study power of 80% in the most important

216 secondary endpoint, the MFI20 questionnaire. The” What is the statistic test that will be used? Is there any evidence for the expectation of 5-10 MFI units?

Please refer to the yellow section above, there specifics and details of the sample size estimation.

Could you explain what successful acceptance is and how it will be used? “Successful acceptance and compliance are achieved when the acceptance criteria, as defined

228 in the primary endpoint, are met by at least 80% of patients.” What is the acceptance criteria? Do you mean if more than 80% of patients use taVNS for more than 4 hours, then taVNS is feasible?

Yes, we corrected that.

Line 264, it seems that this study includes a titration of intensity. Please state how the investigator will reduce the intensity from the initial setting, for example, with a decrement of 5 units when the patient reports feeling the stimulation. How do you ensure that the Verum -1 group uses the predefined stimulation intensity?

We corrected that – the patients will repeat the procedure to actively downregulate the intensity below threshold.

In study plan, the study started last year. Have any patients been recruited? Is there any preliminary data that can be used to estimate the effect size?

We have not yet performed any analysis as the data is still “sealed” in our Recap-System. It will be analyzed once the study is completed.

3. In conflict of interest, as “MG is founder and advisor of Neuroloop GmbH,” The authors need to state whether this company will manufacture products to reduce fatigue in patients with long-covid syndrome.

This statedment was added.

4. In the discussion section, fatigue is stated as one of the side effects of taVNS in a meta-analysis published in a scientific report. Please discuss whether this side effect will confound the analysis.

The only side effect of taVNS close to fatigue is dizziness, which means vertigo in this context. Fatigue is no known side effect of taVNS.

5. Please verify if cymba conchae, rather than tragus, is most innervated by the auricular branch of the vagus nerve. (https://www.ncbi.nlm.nih.gov/pmc/articles/PMC7083568/). In reference [22], it seems that they also indicates that tragus is partly innervated by the auricular branch of the vagus nerve. The authors might need additional evidence to support stimulation of aVN through tragus, for example, tragus is easier to access.

We corrected this, it was our fault – our system stimulates the cymba conchae which is innervated by the auricular branch of the vagal nerve. We added the respective literature.

Minor comments

1. Abstract, first sentence, I understand this data comes from study. But to provide a background for this protocol, I suggest that the authors state for example, estimated number of patients who is currently having long covid syndrome.

We added a respective link to the WHO, stating that 775 mio. Patients suffered from Covid-19. If only 10% of them get long covid, it is more than 77mio patients.

2. Page 3, line 70, “In addition to fatigue (58%), these patients experience headaches

(44%), anosmia, anxiety (13%), and depression (12%). Please specify the denominator used to calculate these percentages.

We corrected this an just referred to one study with 143 patients.

3. Key message, page 3, is the key message section required to be published in PLOS ONE? Key messages in the manuscript seem to be a shorter abstract. The manuscript without the key messages is complete.

We deleted them.

4. Please rephrase sentences 81 – 83; for example, ‘fatigue can continue to affect these patients months after recovery from the infection.’

We rephrased the sentence.

5. The use of abbreviations needs to be consistent. For example, line 115 is transcutaneous VNS taVNS?

There are other transcutaneous systems stimulating the vagle nerve directly through the skin. We change “transcutaneous” to “non-invasive” VNS to make this more clear.

6. What does Verum stand for?

“Verum” is the latin word for “true”, which means in this context “true stimulation group”.

7. Change references into English. Are the journal name and publication date in Germany?

Yes, sorry, we corrected that.

8. Optional: provide instruction to the three groups of patients as supplementary material.

The patients receive the official manual of the stimulator and a verbal instruction of the system. The stimulator is very easy to use so the patients don’t really need any written instruction.

9. Lin 240, please revise “the effect between 239 suprathreshold and subthreshold

10. 240 stimulation will be investigated using two Verum groups.”

We rephrased this sentence.

11. What is CE in “CE-certi

---

## [Decision Letter · Decision Letter 1]

5 Nov 2024

PONE-D-23-38498R1COVIVA: Effect of transcutaneous auricular vagal nerve stimulation on fatigue-syndrome in patients with Long Covid – a placebo-controlled pilot study protocolPLOS ONE

Dear Dr. Gierthmuehlen,

Thank you for submitting your manuscript to PLOS ONE. After careful consideration, we feel that it has merit but does not fully meet PLOS ONE’s publication criteria as it currently stands. Therefore, we invite you to submit a revised version of the manuscript that addresses the points raised during the review process.

The manuscript has significantly improved, and I appreciate the authors' efforts. However, the statistical section still requires further revision before the article can be accepted.

We look forward to receiving your revised manuscript.

Kind regards,

Mohammadreza Pourahmadi, PT, Ph.D., Postdoctoral Fellow

Academic Editor

PLOS ONE

Journal Requirements:

Reviewers' comments:

Reviewer's Responses to Questions

**Comments to the Author**

1. Does the manuscript provide a valid rationale for the proposed study, with clearly identified and justified research questions?

Reviewer #2: Yes

Reviewer #3: Yes

2. Is the protocol technically sound and planned in a manner that will lead to a meaningful outcome and allow testing the stated hypotheses?

Reviewer #2: Partly

Reviewer #3: Yes

3. Is the methodology feasible and described in sufficient detail to allow the work to be replicable?

Reviewer #2: Yes

Reviewer #3: Yes

4. Have the authors described where all data underlying the findings will be made available when the study is complete?

Reviewer #2: Yes

Reviewer #3: Yes

5. Is the manuscript presented in an intelligible fashion and written in standard English?

Reviewer #2: Yes

Reviewer #3: Yes

6. Review Comments to the Author

You may also provide optional suggestions and comments to authors that they might find helpful in planning their study.

Reviewer #2: The manuscript, which has been improved in the current revised version, describes a study protocol investigating the feasibility and effect of applying taVNS in Long Covid patients. Most of my previous questions have been addressed. However, the section on statistical analysis, including the sample size calculation and hypothesis testing, requires further revision. Below are my detailed comments for review.

1. Line 199, Data analysis (hypothesis testing) should be separated from sample size calculation.

2. Line 211, the word ‘upgraded’ is ambiguous.

3. Line 206, I believe that general readers will understand if p-value < threshold means that the null hypothesis is rejected. If the authors want to make this statement specific, please correct ‘test value’ to ‘p-value’ and state explicitly what the null hypothesis is. Alsom, Line 203, will the author perform two tests on the primary outcome and ranked judgment scales?

4. Line 209, the treatment groups will be descriptively compared with the placebo group? This contradicts the previous statement that statistical tests will be used to compare (the primary outcome) between two groups. What are the two groups?

5. P219, I assume the calculation has been performed in R and will not change in the future.

6. Line 220, ‘analyzed descriptively using the same statistical test,’ is not clear. Does it mean the conclusion of this proposed study will not be drawn based on BDI, SF-36, and PCS-score? Are these comparisons only for generating hypotheses for future clinical trials? The authors should be cautious when making this statement as it will restrict what will be included in future publications.

7. Line 254, ‘Furthermore, no adverse effects associated with stimulation 255 durations exceeding 4 h/day will be reported.’ Do the authors only consider adverse effects occurring within the 4 hours after the stimulation?

Reviewer #3: Line numbers in track-changes revised manuscript

Line 52: What does “implantation of a non-functional electrode” refer to? Implantation implies invasive device (yours is not).

Line 106: Would consider addressing the disparity between your statement “Existing research indicates that fatigue is partly triggered by increased levels of pro-inflammatory hormones known as cytokines in the blood, while its severity seems to correlate with the levels of these cytokines in the initial phase of the disease” with the fact that your enrolled patients are not in the acute phase of the disease. This implies prevention of chronic-fatigue would be better tackled by initiating therapy early in the acute setting of the disease to take advantage of the anti-inflammatory effect of taVNS.

Line 286: Do you mean that adverse events have not been previously reported? Would consider reviewing and addressing concerns raised by a recent meta-analysis looking at the safety profile of taVNS that stated “Repeated sessions and sessions lasting 60 min or more are also shown to be more likely to lead to AEs” (PMID 36543841). In particular, this should be addressed for patients that could over deliver even longer stimulation times than described in the protocol.

Line 306: type “able”

Line 360: Would describe this an optional cross-over design

Line 370: It is a little misleading to state “Several studies have shown that healthy patients undergo acute taVNS to investigate scientific questions in behavioral research [55–57]. None of the existing studies on this topic have reported any relevant side effects.” While generally well-tolerated, there are reported adverse events for taVNS, particularly for longer stimulation durations as you mention several lines later. Your enrolled patients are also not necessarily “healthy”, as you exclude some severe physical and mental comorbidities, but patients may have other confounding diagnoses that could impact the safety profile of your study.

As previous research has indicated VNS increases arousal, and this may certainly combat symptoms of fatigue, have you considered any directions to participants for timing of the intervention. IE performed during early hours of the day rather than just before bed?

7. PLOS authors have the option to publish the peer review history of their article (what does this mean? ). If published, this will include your full peer review and any attached files.

**Do you want your identity to be public for this peer review?** For information about this choice, including consent withdrawal, please see our Privacy Policy .

Reviewer #2: **Yes: ** Gansheng Tan

Reviewer #3: **Yes: ** Anna L Huguenard

---

## [Author Response · Author response to Decision Letter 1]

28 Nov 2024

We thank the reviewers and the editor for their helpful comments

Reviewer #2: The manuscript, which has been improved in the current revised version, describes a study protocol investigating the feasibility and effect of applying taVNS in Long Covid patients. Most of my previous questions have been addressed. However, the section on statistical analysis, including the sample size calculation and hypothesis testing, requires further revision. Below are my detailed comments for review.

1. Line 199, Data analysis (hypothesis testing) should be separated from sample size calculation.

Done.

2. Line 211, the word ‘upgraded’ is ambiguous.

We corrected that.

3. Line 206, I believe that general readers will understand if p-value < threshold means that the null hypothesis is rejected. If the authors want to make this statement specific, please correct ‘test value’ to ‘p-value’ and state explicitly what the null hypothesis is. Alsom, Line 203, will the author perform two tests on the primary outcome and ranked judgment scales?

The test statistic will evaluate the difference between the three groups in terms of binary data (primary outcome is an incidence rate with patients who exceed a usage-time-threshold) using the Fisher-test (exact test), to obtain a two-sided p-value. Additionally, the two treatment groups will be descriptively compared with the placebo group, and between themselves.

Since three tests (each group compared with each other) are performed and the threshold for the p-value was set at an overall (studywise) alpha of 0.05, then three tests result in an Bonferroni-adjusted (comparisonwise) threshold of p / 3 = 0.017. p-values below this threshold denote a statistical significant difference.

4. Line 209, the treatment groups will be descriptively compared with the placebo group? This contradicts the previous statement that statistical tests will be used to compare (the primary outcome) between two groups. What are the two groups?

There are three groups in total, two treatment groups and a placebo group. 5. P219, I assume the calculation has been performed in R and will not change in the future.

That is correct.

6. Line 220, ‘analyzed descriptively using the same statistical test,’ is not clear. Does it mean the conclusion of this proposed study will not be drawn based on BDI, SF-36, and PCS-score? Are these comparisons only for generating hypotheses for future clinical trials? The authors should be cautious when making this statement as it will restrict what will be included in future publications.

BDI, SF36 and PCSS are secondary measures which will be analyzed descriptively. Since those parameters are continous rank-analses of variance (Kruskal wallis tests) will be used.

7. Line 254, ‘Furthermore, no adverse effects associated with stimulation 255 durations exceeding 4 h/day will be reported.’ Do the authors only consider adverse effects occurring within the 4 hours after the stimulation?

You are right – this sentence was modified in the wrong way, I apologize. I wanted to say that in literature there is no evidence for negative effects in case the stimulation is performed for more than 4 hours.

Reviewer #3: Line numbers in track-changes revised manuscript

Done

Line 52: What does “implantation of a non-functional electrode” refer to? Implantation implies invasive device (yours is not).

That is correct. We changed it to “application”.

Line 106: Would consider addressing the disparity between your statement “Existing research indicates that fatigue is partly triggered by increased levels of pro-inflammatory hormones known as cytokines in the blood, while its severity seems to correlate with the levels of these cytokines in the initial phase of the disease” with the fact that your enrolled patients are not in the acute phase of the disease. This implies prevention of chronic-fatigue would be better tackled by initiating therapy early in the acute setting of the disease to take advantage of the anti-inflammatory effect of taVNS.

We modified the sentence that the development of fatigue is linked to cytokines, while its severity seems to correlate with the initial cytokine levels.

Line 286: Do you mean that adverse events have not been previously reported? Would consider reviewing and addressing concerns raised by a recent meta-analysis looking at the safety profile of taVNS that stated “Repeated sessions and sessions lasting 60 min or more are also shown to be more likely to lead to AEs” (PMID 36543841). In particular, this should be addressed for patients that could over deliver even longer stimulation times than described in the protocol.

Indeed. We corrected the sentence and added the literature.

Line 306: type “able”

We corrected this.

Line 360: Would describe this an optional cross-over design

Done

Line 370: It is a little misleading to state “Several studies have shown that healthy patients undergo acute taVNS to investigate scientific questions in behavioral research [55–57]. None of the existing studies on this topic have reported any relevant side effects.” While generally well-tolerated, there are reported adverse events for taVNS, particularly for longer stimulation durations as you mention several lines later. Your enrolled patients are also not necessarily “healthy”, as you exclude some severe physical and mental comorbidities, but patients may have other confounding diagnoses that could impact the safety profile of your study.

We corrected that paragraph.

As previous research has indicated VNS increases arousal, and this may certainly combat symptoms of fatigue, have you considered any directions to participants for timing of the intervention. IE performed during early hours of the day rather than just before bed?

Since other studies also do not impose such requirements and we are aiming for maximum patient compliance, we did not specify a fixed schedule for when the stimulation should take place.

---

## [Decision Letter · Decision Letter 2]

26 Feb 2025

PONE-D-23-38498R2COVIVA: Effect of transcutaneous auricular vagal nerve stimulation on fatigue-syndrome in patients with Long Covid – a placebo-controlled pilot study protocolPLOS ONE

Dear Dr. Gierthmuehlen,

Thank you for submitting your manuscript to PLOS ONE. After careful consideration, we feel that it has merit but does not fully meet PLOS ONE’s publication criteria as it currently stands. Therefore, we invite you to submit a revised version of the manuscript that addresses the points raised during the review process.

We look forward to receiving your revised manuscript.

Kind regards,

Usman Ghafoor

Academic Editor

PLOS ONE

Journal Requirements:

Reviewers' comments:

Reviewer's Responses to Questions

**Comments to the Author**

1. Does the manuscript provide a valid rationale for the proposed study, with clearly identified and justified research questions?

Reviewer #2: Yes

Reviewer #4: Yes

Reviewer #5: Yes

2. Is the protocol technically sound and planned in a manner that will lead to a meaningful outcome and allow testing the stated hypotheses?

Reviewer #2: Yes

Reviewer #4: Yes

Reviewer #5: Yes

3. Is the methodology feasible and described in sufficient detail to allow the work to be replicable?

Reviewer #2: Yes

Reviewer #4: Yes

Reviewer #5: Yes

4. Have the authors described where all data underlying the findings will be made available when the study is complete?

Reviewer #2: No

Reviewer #4: Yes

Reviewer #5: Yes

5. Is the manuscript presented in an intelligible fashion and written in standard English?

Reviewer #2: Yes

Reviewer #4: Yes

Reviewer #5: Yes

6. Review Comments to the Author

You may also provide optional suggestions and comments to authors that they might find helpful in planning their study.

Reviewer #2: The revision is satisfactory, and I recommend the manuscript for publication. One of the reviewer evaluation criteria is "Have the authors described where all data underlying the findings will be made available when the study is complete?" The authors can consider adding such a statement. I noticed that there are two sections titled ‘Study Plan’ in the manuscript. These changes do not require another round of review.

Reviewer #4: This is a short trial designed to study effect of TAV nerve stimulation on fatigue-syndrome in patients with long Covid. I have some comments to improve the presentation of the protocol.

1. In line: 214: It should be PCS score. The word “score” is missing.

2. In line, 166 and 167 say that primary endpoint is measure by the average stimulation duration (which is a continuous variable). On the other hand, line 297 says that primary outcome is an incidence rate with patients who exceed a usage-time-threshold (which is a binary categorical variable). The definitions need to be consistent. See also the definition of primary end point in the discussion section of the abstract.

3. In line 218, which test statistics was used to calculate the sample size of 36 needs to be mentioned.

4. In line 223, It is better to mention that “other” secondary measures BDI, SF36 and PCSS……. This is because the first secondary measure MFI20 is already mentioned.

5. Sample size calculation section only has got the reference for sample size calculation ideas but not the exact sample size calculation for the trial. On the other hand, Data analysis section has got sample size calculation, statistical analysis as well as definition of end points. It would be better if clear definitions of primary and secondary end points are moved to the end points section. Similarly, sample size calculation and statistical analysis sections separated for clarity of presentation.

Reviewer #5: The manuscript investigates the feasibility and efficacy of transcutaneous auricular vagal nerve stimulation (taVNS) for reducing fatigue in long Covid patients. A placebo-controlled pilot study with 45 participants will compare above-threshold, below-threshold, and sham stimulation over 4 weeks. The primary endpoint is adherence to the protocol, while secondary outcomes include fatigue severity, quality of life, and depressive symptoms, assessed via questionnaires. The reviewer has the following concerns.

1. The manuscript describes the use of Fisher’s exact test and Kruskal-Wallis tests, but the rationale for selecting these statistical methods over alternatives should be explicitly justified, particularly in comparing the three groups.

2. The sample size calculation is based on prior taVNS studies, but there is no clear power analysis for the primary endpoint. Clarify whether a sensitivity analysis was conducted to account for expected dropout rates and compliance variability.

3. The sham control uses a non-functional electrode, but prior research suggests that even subthreshold stimulation may induce physiological effects. A stronger justification for this approach or a discussion on potential placebo effects and blinding validation would strengthen the study design.

4. The protocol allows up to 4 hours of daily taVNS, yet studies indicate that prolonged stimulation (>60 minutes) may lead to side effects such as headaches or nausea. The authors must comment on this.

5. Since taVNS influences arousal and autonomic function, the timing of stimulation (e.g., morning vs. evening) could impact outcomes. Clarify whether stimulation timing should be standardized or at least documented to analyze its effects on fatigue and QoL.

7. PLOS authors have the option to publish the peer review history of their article (what does this mean? ). If published, this will include your full peer review and any attached files.

**Do you want your identity to be public for this peer review?** For information about this choice, including consent withdrawal, please see our Privacy Policy .

Reviewer #2: **Yes: ** Gansheng Tan

Reviewer #4: **Yes: ** Dr Shah-Jalal Sarker

Reviewer #5: No

---

## [Author Response · Author response to Decision Letter 2]

11 Mar 2025

Comments to the reviewers

We thank the reviewers for their valuable comments.

Reviewer #2:

The revision is satisfactory, and I recommend the manuscript for publication. One of the reviewer evaluation criteria is "Have the authors described where all data underlying the findings will be made available when the study is complete?" The authors can consider adding such a statement. I noticed that there are two sections titled ‘Study Plan’ in the manuscript. These changes do not require another round of review.

We changed the second “Study plan” to “Study procedure”. We also added a statement that the results will be published in peer reviewed journals.

Reviewer #4:

1. In line: 214: It should be PCS score. The word “score” is missing.

We corrected this accordingly.

2. In line, 166 and 167 say that primary endpoint is measure by the average stimulation duration (which is a continuous variable). On the other hand, line 297 says that primary outcome is an incidence rate with patients who exceed a usage-time-threshold (which is a binary categorical variable). The definitions need to be consistent. See also the definition of primary end point in the discussion section of the abstract.

This is correct, we changed that.

3. In line 218, which test statistics was used to calculate the sample size of 36 needs to be mentioned.

We added the respective information.

4. In line 223, It is better to mention that “other” secondary measures BDI, SF36 and PCSS……. This is because the first secondary measure MFI20 is already mentioned.

Corrected.

5. Sample size calculation section only has got the reference for sample size calculation ideas but not the exact sample size calculation for the trial. On the other hand, Data analysis section has got sample size calculation, statistical analysis as well as definition of end points. It would be better if clear definitions of primary and secondary end points are moved to the end points section. Similarly, sample size calculation and statistical analysis sections separated for clarity of presentation.

We corrected that, too.

Reviewer #5:

1. The manuscript describes the use of Fisher’s exact test and Kruskal-Wallis tests, but the rationale for selecting these statistical methods over alternatives should be explicitly justified, particularly in comparing the three groups.

This was added.

2. The sample size calculation is based on prior taVNS studies, but there is no clear power analysis for the primary endpoint. Clarify whether a sensitivity analysis was conducted to account for expected dropout rates and compliance variability.

The power analysis is now in line 208. The drop-out-rate was estimated with the numbers of Redgrave et. Al. 2018 in mind, stating that throughout all studies in the meta-analysis around 3% of all participants drop out. In patients with long-covid we estimated this number a bit higher.

3. The sham control uses a non-functional electrode, but prior research suggests that even subthreshold stimulation may induce physiological effects. A stronger justification for this approach or a discussion on potential placebo effects and blinding validation would strengthen the study design.

We have three groups: Sham, Verum-1 and Verum-2. Verum-2 is standard stimulation. Verum-1 is stimulation below the threshold, but still a stimulation. Sham is no stimulation at all, guaranteed by a non-functional plastic-electrode with no stimulation capability. This is mentioned in line 298.

4. The protocol allows up to 4 hours of daily taVNS, yet studies indicate that prolonged stimulation (>60 minutes) may lead to side effects such as headaches or nausea. The authors must comment on this.

We added a respective sentence. However, the stimulator we use has a CE-certificate as a medical product in Germany and is certified for recommended use of 4h/day. Both software and integrated timer are calibrated to this recommendation.

5. Since taVNS influences arousal and autonomic function, the timing of stimulation (e.g., morning vs. evening) could impact outcomes. Clarify whether stimulation timing should be standardized or at least documented to analyze its effects on fatigue and QoL.

Thank you, this is an interesting point, indeed. To our knowledge, no study has so far addressed this aspect, also the manual of the stimulator does not recommend any specific time. However, the time of stimulation is logged both by the app and the patient’s stimulation diary and will consider analyzing this aspect.

---

## [Decision Letter · Decision Letter 3]

2 Apr 2025

COVIVA: Effect of transcutaneous auricular vagal nerve stimulation on fatigue-syndrome in patients with Long Covid – a placebo-controlled pilot study protocol

PONE-D-23-38498R3

Dear Dr. Gierthmuehlen,

We’re pleased to inform you that your manuscript has been judged scientifically suitable for publication and will be formally accepted for publication once it meets all outstanding technical requirements.

Kind regards,

Usman Ghafoor

Academic Editor

PLOS ONE

Additional Editor Comments (optional):

The authors have addressed the comments.

Reviewers' comments:

Reviewer's Responses to Questions

**Comments to the Author**

1. Does the manuscript provide a valid rationale for the proposed study, with clearly identified and justified research questions?

Reviewer #4: Yes

Reviewer #5: Yes

2. Is the protocol technically sound and planned in a manner that will lead to a meaningful outcome and allow testing the stated hypotheses?

Reviewer #4: Yes

Reviewer #5: Yes

3. Is the methodology feasible and described in sufficient detail to allow the work to be replicable?

Reviewer #4: Yes

Reviewer #5: Yes

4. Have the authors described where all data underlying the findings will be made available when the study is complete?

Reviewer #4: Yes

Reviewer #5: Yes

5. Is the manuscript presented in an intelligible fashion and written in standard English?

Reviewer #4: Yes

Reviewer #5: Yes

6. Review Comments to the Author

You may also provide optional suggestions and comments to authors that they might find helpful in planning their study.

Reviewer #4: The authors have applied all of my comments for improvement and hence it may be accepted for publication.

Reviewer #5: The authors have addressed all my concerns comprehensively. The manuscript is now suitable for publication and I recommend acceptance without further revisions.

7. PLOS authors have the option to publish the peer review history of their article (what does this mean? ). If published, this will include your full peer review and any attached files.

**Do you want your identity to be public for this peer review?** For information about this choice, including consent withdrawal, please see our Privacy Policy .

Reviewer #4: **Yes: ** Dr Shah-Jalal Sarker

Reviewer #5: No

---

## [Editor Report · Acceptance letter]

PONE-D-23-38498R3

PLOS ONE

Dear Dr. Gierthmuehlen,

I'm pleased to inform you that your manuscript has been deemed suitable for publication in PLOS ONE. Congratulations! Your manuscript is now being handed over to our production team.

Kind regards,

on behalf of

Dr. Usman Ghafoor

Academic Editor

PLOS ONE